# Dietary Antimicrobial Peptides Improve Intestinal Function, Microbial Composition and Oxidative Stress Induced by *Aeromonas hydrophila* in Pengze Crucian Carp (*Carassius auratus* var. *Pengze*)

**DOI:** 10.3390/antiox11091756

**Published:** 2022-09-06

**Authors:** Shaodan Wang, Shulin Liu, Chong Wang, Bin Ye, Liqun Lv, Qiao Ye, Shaolin Xie, Guocheng Hu, Jixing Zou

**Affiliations:** 1Joint Laboratory of Guangdong Province and Hong Kong Region on Marine Bioresource Conservation and Exploitation, College of Marine Sciences, South China Agricultural University, Guangzhou 510642, China; 2Guangdong Laboratory for Lingnan Modern Agriculture, South China Agricultural University, Guangzhou 510642, China; 3National Pathogen Collection Center for Aquatic Animals, Key Laboratory of Freshwater Fishery Germplasm Resources, Shanghai Ocean University, Shanghai 201306, China; 4School of Life Sciences, Huizhou University, Huizhou 516007, China; 5State Environmental Protection Key Laboratory of Environmental Pollution Health Risk Assessment, South China Institute of Environmental Sciences, Ministry of Ecology and Environment, Guangzhou 510655, China

**Keywords:** antimicrobial peptides, oxidative stress, intestinal morphology, intestinal microbes, *Carassius auratus* var. *Pengze*

## Abstract

There is increasing evidence for the potential use of antimicrobial peptides as dietary supplements and antibiotic substitutes. In this study, we analyzed the differential effects of varying levels of antimicrobial peptides on the intestinal function and intestinal microbial and disease resistance of Pengze crucian carp. Approximately 630 experimental fishes were randomized in the control group (G0: 0 mg/kg) and in five groups supplemented with different doses of AMPs (G1: 100 mg/kg, G2: 200 mg/kg, G3: 400 mg/kg, G4: 800 mg/kg, and G5: 1600 mg/kg) and were fed for ten weeks. Three replicates per group of 35 fish were performed. The results showed that AMPs promoted intestinal villus development and increased intestinal muscular thickness (*p* < 0.05) and goblet cell abundance. The enzymatic activities of all groups supplemented with AMPs were effectively improved. AMP supplementation significantly enhanced the activities of antioxidant enzymes and digestive enzymes in the intestines of G3 animals (*p* < 0.05). Compared with G0 animals, AMP-supplemented animals regulated the expression of intestinal immune-related genes and exhibited significant differences in the G3 animal group (*p* < 0.05). The abundance of intestinal *Firmicutes* and *Bacteroidetes* increased in the AMP-supplemented groups, but the *Firmicutes/Bacteroidetes* ratio was lower than that in the G0 group. AMP supplementation also decreased the abundance of *Fusobacterium* while increasing the proportion of *Actinobacteria* (*p* < 0.05). After *Aeromonas hydrophila* infection, the expression levels of anti-inflammatory factors in the intestinal tract of G3 animals were significantly upregulated, and the level of the proinflammatory factor was decreased (*p* < 0.05). The intestinal *Cetobacterium* levels of G3 animals were significantly increased (*p* < 0.01), while the *Proteobacteria* levels were decreased, and the intestinal goblet cell proliferation was significantly lower than that of G0 animals (*p* < 0.05). This indicates that groups supplemented with AMPs have better disease resistance than the G0 group and can rapidly reduce the adverse effects caused by inflammatory response. Taken together, the present results suggest that AMP supplementation can improve intestinal function and intestinal microbial and pathogen resistance in Pengze crucian carp.

## 1. Introduction

At the beginning of the 20th century, with the discovery and promotion of penicillin, antibiotics were widely used in many fields, including clinical treatment to prevent bacterial infection and in production processes as an additive to promote yield [1]. Alarmingly, the overuse of antibiotics in clinical treatment has favored, in recent decades, an increase in antibiotic-resistant pathogens [2]. What followed was the emergence of multidrug-resistant and extensively drug-resistant bacteria [3]. Today, the concept of exploiting antimicrobial natural sources as novel antibacterial therapeutics represents a paradigm shift [4].

Antimicrobial peptides (AMPs) are small-molecule peptides that widely exist in plants [5], animals [6], and other ecological species [7]. They consist of dozens of amino acid residues with broad-spectrum antibacterial properties and low drug resistance [8]. Natural AMPs produced by the body are regulated by the TLR signaling pathway and have broad-spectrum activities against bacteria, fungi, viruses and tumor cells and mediate apoptosis and immune regulation [4]. In addition, the antibacterial activity of AMPs is closely related to their special secondary structures, α-helical linear structures and β-sheet circular structures. It is worth noting that an intrinsic component of anti-AMP resistance is still found in some bacteria, albeit at low levels [9]. Therefore, in the research and application of AMPs, the use of combination therapy can not only improve the antibacterial activity but also reduce the generation of drug resistance [10,11].

Further expanding the research on this subject, animal husbandry has also begun to explore the feasibility of using AMPs to replace antibiotics. Studies have indicated that AMPs improve gut barrier function and immunity in weanling pigs [12]. It was speculated that AMPs could reduce the number of intestinal pathogenic bacteria, thereby improving dietary function and intestinal immunity [13]. AMPs were found to promote growth, reduce diarrhea rates, and affect serum parameters in a piglet model of *E. coli* infection [14]. Among husbandry industries, the aquaculture industry has a much higher probability of occurrence and spread of various bacterial and parasitic diseases than other industries due to the characteristics of having a high-density farming mode and water environment transmission. Investigation revealed that *Aeromonas* [15], *Streptococcus* [16], *Vibrio* [17], and other pathogenic bacteria caused the death of a large number of aquatic organisms. In the process, antibiotics and other drugs are widely and uncontrollably used, resulting in the prevalence of serious levels of antibiotic residues in the environment, thus triggering the emergence of drug-resistant bacteria and environmental antibiotic resistance genes (ARGs). By feeding *Epinephelus coioides* a diet supplemented with AMPs, antioxidant capacity and resistance to pathogenic bacteria can be enhanced [18]. Similar results were found in zebrafish (*Danio rerio*), and AMPs increased immune gene expression and the feed conversion ratio [19]. 

Further exploration found that AMPs could improve the abundance and composition of the intestinal microbiota in fish [20,21]. In addition, AMPs were found to significantly reduce the counts of *Escherichia coli* and significantly increase the counts of lactic acid bacteria and *Bifidobacterium* sp. in the intestine of broilers [22,23]. In ruminants, AMPs significantly increased the number and abundance of beneficial bacterial genera in the rumens of juvenile goats and promoted growth and intestinal digestive enzyme activity [24]. Studies have shown that traditional antibiotics destroy the composition of intestinal flora and damage intestinal barrier function, and AMPs have an absolute advantage in this regard [25].

Existing research mainly focuses on exploring the effects of AMPs on mammals, and there is a lack of research data available on aquatic organisms. Most of the research content only includes the effect of AMPs on antioxidant capacity, immunity, and growth performance of the experimental subjects. This study investigated the effects of AMPs on intestinal barrier function, disease resistance and intestinal microbial diversity by supplementing the diet of Pengze crucian carp, aiming to improve some references for antimicrobial peptide research.

## 2. Materials and Methods

### 2.1. Ethics Statement

All procedures involving experimental fish were carried out following the regulations of the animal care and use committee and in accordance with the South China Agricultural University’s Guidelines for Experimental Animals (identification code: 20200315; date of approval: 15 March 2020).

### 2.2. Experimental Fish, Diets, and Experimental Design

The immature Pengze crucian carp (*Carassius auratus* var. *Pengze*) were supplied by the Panyu Agricultural Research Institute (Guangzhou, China). Experimental fish were sterilized using saline water and then acclimatized in a restricted flow and temperature recirculating aquaculture system for 2 weeks and were trained to the diet and breeding environment. Experimental diets and AMPs (antimicrobial peptides C and P) are referred to in a previous study [26], and the composition of the diet is shown in Table 1. The AMPs-C (molecular weight 6 kDa separated by mass spectrometry) was obtained through mixed culture fermentation with E. Coli and a commensal bacterium, isolated from the intestine of free-range chickens. The AMPs-P (molecular weight 5 kDa separated by mass spectrometry) was exocrine peptide of Bacillus amyloliquefaciens, which was separated from the intestines of pigs. The AMPs were mixed according to the ratio of 100,000 and 25,000 AU/g, respectively. Groups were divided based on AMP supplementation doses and were as follows: control, G0 (0 mg/kg), G1 (100 mg/kg), G2 (200 mg/kg), G3 (400 mg/kg), G4 (800 mg/kg) and G5 (1600 mg/kg). Experiments were run in three parallel groups for each group. Fish of similar size were chosen and weighed for the experiment, and approximately 630 experimental fish (3.0 ± 0.05 g) were randomized to sterilized glass tanks in groups of 35. The fish were fed feed rations (3–5 percent of body weight) twice a day (8:30 and 17:30) for ten weeks. The environmental conditions were maintained as follows: a temperature of 26–28 °C, dissolved oxygen (DO) levels of 5.5–6.0 mg/L, and a pH of 7.0–8.0.

### 2.3. Sample Collection

After the trial, the feeding was halted for 24 h and three sample fishes were collected from each tank. The fish were anesthetized with MS-222 at a dosage of 100 mg/L (Sigma, St. Louis, MO, USA). The intestines were collected and refrigerated at an ultralow temperature of −80 °C. In addition, 3 fish were randomly selected from each tank, and the foregut, midgut and hindgut were taken for paraffin sectioning.

### 2.4. Intestinal Morphological Analysis

The morphology of intestinal tissue was assessed using paraffin slices. Fresh tissues were soaked for one week in a 4% paraformaldehyde solution (Servicebio Inc., Wuhan, China) before being dehydrated, paraffin-embedded, sectioned (5–10 um), and stained (hematoxylin and eosin, H&E). A light microscope was used to take photographs and Image-Pro Plus 6.0 to collect data.

### 2.5. Biochemical Parameter Analysis

Frozen intestine samples were thawed on ice and weighed (50–100 mg) before being homogenized with sterile ice-cold saline (0.9% NaCl) to produce a 10% tissue homogenate (*w*/*v*: 1/9). This was followed by centrifugation for 10 min (4 °C, 3500 rpm), where the supernatant was aliquoted and maintained at an ultra-low temperature. Antioxidant enzymes such as total antioxidant capacity (T-AOC), superoxide dismutase (SOD), and malondialdehyde (MDA) were assayed using relevant commercial kits (Nanjing Jiancheng, Bioengineering Institute, Nanjing, China), as were digestive enzymes such as alpha-chymotrypsin (α-chmo), alpha-amylase (α-Ams), and lipases (Lip). Protein content was measured in the samples to assess relevant enzymatic activity.

### 2.6. Gene Expression Analysis

The mRNA levels of toll-like receptor 4 (*TLR-4*), myeloid differentiation primary response gene 88 (*MYD88*), tumor necrosis factor α (*TNF-α*), interleukin 10 (*IL-10*), and interleukin 11 (*IL-11*) in the intestine were assessed with real-time quantitative PCR (RT–qPCR). The isolation of total intestine RNA, reverse-transcription and RT-qPCR were conducted using the following kits and in accordance with the manufacturer’s instructions: TRIzol Reagent (Takara, Beijing, China), PrimeScript™ RT reagent Kit (Takara) and SYBR Green Supermix (Takara), respectively. Table 2 lists the relevant primers (Sangon Biotech, Shanghai, China). Results are reflected as changes in relative expression standardized with *β-actin* using the 2^−ΔΔCt^ method.

### 2.7. 16S Sequencing and Intestinal Microbial Analysis

Microbial genomic DNA was extracted from all samples (*n* = 3) using a FastDNA^®^ SPIN Kit for Soil (MP Biomedicals, Shanghai, China) following the manufacturer’s instructions. The library encompassing the V3–V4 area of the 16S rDNA gene was PCR-amplified with primers (338F: 5′-TCCTACGGGAGGCAGCAG-3′ and 806R: 5′-GGACTACHVGGGTWTCTAAT-3′) and spliced paired-end reads were used to obtain the original spliced sequence (raw contigs). The raw Illumina sequences were obtained from the Illumina MiSeq platform (Illumina, San Diego, CA, USA). Using FLASH and Trimmomatic, low-quality sequences were removed from the raw fastq data [27]. Uchime [28], categorized and eliminated all chimeric sequences, while UPARSE [29] grouped operational taxonomic units (OTUs) using a 97% similarity cutoff.

The RDP classifier algorithm (http://rdp.cme.msu.edu/, accessed on 8 November 2019) was used for classification of the 16S rRNA gene sequence. Alpha diversity analysis included Sobs, Shannon, Simpson, Chao1, ACE, and coverage indices [30]. Beta diversity analysis included principal component analysis (PCA) and principal coordinate analysis (PCoA) [31]. PICRUSt was utilized to investigate variations in gene function across groups [32].

### 2.8. Aeromonas Hydrophila Infection Analysis

#### 2.8.1. Vitro Antibacterial Test

To extract the active components, 0.16 g of AMPs was dissolved in 1 mL of 65% ethanol, then the mixture was centrifuged to separate the supernatant. With 65% ethanol, AMPs supernatant was diluted to create detection solutions at concentrations of 160 mg/mL, 16 mg/mL, 1.6 mg/mL, 0.16 mg/mL, and 0.016 mg/mL. Using the filter paper approach, *Aeromonas hydrophila* in vitro antibacterial detection was carried out on AMPs. A sterile swab was used to uniformly distribute the ten-fold diluted log-phase bacterial stock solution (10^7^/mL) on LB agar medium. The 4 mm sterilized filter paper was placed on the *Aeromonas hydrophila*-inoculated plate’s surface. An amount of 10 uL of each of the 5 different concentrations of AMP detection solutions and 65% ethanol were dropped on the filter paper as a positive control and a negative control, respectively. After incubation at 37 °C for 10 h, the diameter of the inhibition zone was measured.

#### 2.8.2. Vivo infection Test

A total of 15 fish were randomly chosen out of each experimental group and inoculated with a certain dosage of *Aeromonas hydrophila* liquid (donated by Foshan Academy of Sciences). Primary bacterial liquid was prepared as described previously [33]. Each trial fish was inoculated intraperitoneally with 50 μL of *Aeromonas hydrophila* liquid (8.56 × 107 CFU/mL) diluted with PBS (Phosphate-Buffered Saline, pH 7.0–7.2). In addition, 15 fish selected from the G0 group were injected with PBS, serving as the control group. After 48 h, tissues were collected and examined by gene expression, as described above.

The stock solution of *Aeromonas hydrophila* was diluted to different concentrations using the step-by-step dilution method, and the fish in the control group were selected for the LC50 experiment. Plate counts of the diluted bacterial solutions with different concentrations were used to determine the actual concentration.

### 2.9. Statistical Analysis

Excel 2016, Image-Pro Plus 6.0 (Media Cybernetics Inc., Rockville, MD, USA), GraphPad Prism 7.0 (GraphPad Inc., San Diego, CA, USA), and SPSS 20 (SPSS Inc., Chicago, IL, USA) were used to perform statistical analyses and for graphing the data. One-way analysis of variance (ANOVA) and Duncan’s method for multiple comparisons (*p* < 0.05) were used to identify the differences in the data. Mothur software was used to analyze the alpha and beta diversity. In this study, other assessments of intestinal microbiota were carried out using the R package software.

## 3. Results

### 3.1. AMPs Alter Intestinal Morphology

Appendix A shows the villus height, villus width, muscular thickness, goblet cell number (per 100 μm), and villus count in the foregut, midgut, and hindgut of the G0 and AMP-supplemented groups (Appendix A). The results show that a certain level of supplemental AMPs can increase the length, width and count of intestinal villi and increase the abundance of goblet cells. The foregut, midgut and hindgut showed the highest improvement in G3 animals compared to G0 animals. In G3 animals, AMPs significantly increased muscular thickness (*p* < 0.05), but there were no significant differences in the midgut (*p* > 0.05). Additionally, they enhanced the abundance of goblet cells.

### 3.2. AMPs Affect Intestinal Biochemical Parameters

The intestinal enzyme activities are shown in Table 3. The results demonstrate that different doses of AMP supplementation can boost antioxidant capacity in the intestine. Compared with G0 animals, the activities of T-AOC and SOD in G3 animals were substantially increased (*p* < 0.05), and both values peaked in the G3 group. The T-AOC activity of G5 animals and the SOD activity of G2 animals were in opposition to those of the other groups but were not statistically different from those of the G0 group (*p* > 0.05). The activity of MDA in the G3 group was substantially downregulated compared with that in the G0 group (*p* < 0.05). AMPs can boost the activity of α-chmo, α-Ams, and Lip in the intestine. Compared with the G0 group, the activities of α-chmo, α-Ams, and Lip were upregulated with the addition of AMPs, and peaked in value in the G3 group (*p* < 0.05). High amounts of AMPs would inhibit the activity of α-chmo, α-Ams, and Lip when compared to the G3 group.

### 3.3. AMPs Regulate Intestinal Immune-Related Gene Expression

Figure 1 shows the relative expression of immune-related genes in the intestine. Compared with the G0 group, AMPs lead to the activation of *TLR-4* (Figure 1A), *MYD88* (Figure 1B), and *TNF-α* (Figure 1C) in the intestine, with the G3 group having the peak value (*p* < 0.05). Other AMP-supplemented groups also showed significant upregulation in some or all *TLR-4*, *MYD88*, and *TNF-α* levels (*p* < 0.05). AMPs lowered the relative expression of *IL-10* (Figure 1D) and *IL-11* (Figure 1E) in all groups except in the G1 group, with the lowest relative expression in the G3 group (*p* < 0.05).

### 3.4. AMPs Supplementation and Intestinal Microbiota Diversity

The intestinal microbial 16S rDNA sequencing built 863182 high-quality sequences, with no significant difference between groups (*p* > 0.05) (Figure 2A). PCA and PCoA were performed at the OTU level to observe the degree of dispersion between groups (Figure 2B,C). A dispersion was observed between groups G3 and G0, but the difference was not significant (Figure 2E,F). The overlapping section of the Venn diagram represents the shared OTUs between different groups, and the nonoverlapping section represents the unique OTUs of that group (Figure 2D). At the OTU level, AMPs altered the alpha diversity index of the intestinal microbiota (Appendix A). Compared with the G0 group, AMPs increased the community diversity and richness of the intestinal microbiota. The Shannon (Figure 2G) and Chao1 (Figure 2J) indices in the AMP-supplemented groups were increased compared with those in the G0 group and reached their peak value in the G3 group (*p* < 0.05). There were no significant differences in the Ace index compared with G0 group, but there was an upward trend observed (Figure 2H). The Simpson index was downregulated in the AMP groups and was significantly different in the G3 group (*p* < 0.01) (Figure 2I). At the phylum level, the predominant bacterial phyla in the intestine were *Fusobacteria, Proteobacteria, Firmicutes, Actinobacteria, Bacteroidetes,* and *Cyanobacteria* (Figure 3A). At the genus level, *Cetobacterium, Aeromonas, ZOR0006, Shewanella, Burkholderia-Caballeronia-Paraburkholderia, Timonella, unclassified_f_Erysipelotrichaceae, Flavobacterium, Gemmobacter*, and *Nocardia* comprised the top ten dominant genera in the intestinal bacterial community (Figure 3B). Compared with the G0 group, the AMP-supplemented groups had a total of six significantly different OTU sets (OTU685, OTU95, OTU425, OTU459, OTU456, and OTU91) at the top 15 levels of OTUs (Figure 3C). By comparing the pairwise differences between different AMP-supplemented groups and the G0 group, we found that the G3 group had the highest difference (Figure 3D–H). Compared with G0, the G3 group exhibited significant downregulation in the numbers of *Fusobacteria* and *Cetobacterium* and significant upregulation in the numbers of *Actinobacteria*, *Flavobacterium* and *Rhodococcus* (Figure 3I,J) (*p* < 0.05). Through statistical analysis of the abundance of the intestinal community, COG function of the AMP-supplemented and G0 groups showed that the enriched functions mainly included energy production and conversion, amino acid transport and metabolism, and general function prediction only, among others (Figure 4B). AMPs increased the relative abundance of these functions (Figure 4A).

### 3.5. AMPs Extracts’ In Vitro Bacteriostatic Efficacy against Aeromonas hydrophila 

*Aeromonas hydrophila* was used as a test subject for the antibacterial activity of AMPs at various doses (Table 4). The outcomes demonstrated that the 1.6 mg/mL concentration of AMPs allowed for the observation of the clear inhibitory zone. AMPs cannot completely disperse on agar plates since they are amphiphilic biomacromolecules. Because of this, the inhibitory zone did not significantly change between high-concentration AMPs (160 mg/mL) and low-concentration AMPs (1.6 mg/mL). There was no evident inhibitory zone at the lower dosages of 0.16 mg/mL and 0.016 mg/mL (Appendix A).

### 3.6. AMPs Improve the Resistance of the Intestinal Ecosystem to Aeromonas hydrophila Infection

#### 3.6.1. AMPs Reduce Intestinal Damage after *Aeromonas hydrophila* Infection

*Aeromonas hydrophila* infection had a significant effect on intestinal goblet cell abundance (Appendix A) (Appendix A). Changes in goblet cell abundance were observed in all infected groups. Intestinal goblet cell abundance was increased in the G0 group compared to the PBS control group, and significant differences were observed in the midgut and hindgut (*p* < 0.05). However, the abundance of goblet cells in the AMP-supplemented group was decreased compared with that in the G0 group (*p* > 0.05), and the number of goblet cells in the G3 group was significantly lower than that in the G0 group (*p* < 0.05).

#### 3.6.2. AMPs Increase Intestinal Immune-Related Gene Expression after *Aeromonas hydrophila* infection

*Aeromonas hydrophila* infection resulted in enhanced immune gene expression in all groups compared to the PBS control group (Figure 5). Compared with the G0 group, the expression of *TLR-4*, *MYD88*, as well as *TNF-α* in the AMP-supplemented groups was lower (Figure 5A–C), and there was a significant difference in the G3 group (*p* < 0.05). Other AMP-supplemented groups also exhibited a downregulation trend of *TLR-4*, *MYD88*, and *TNF-α* levels. In contrast, the relative expression of IL-10 and IL-11 was increased in those groups compared to the G0 group (Figure 5D,E). The expression level in the G3 group had considerably greater levels than the G0 group (*p* < 0.05).

#### 3.6.3. AMPs Ameliorate Intestinal Microbiota Diversity after *Aeromonas hydrophila* Infection

The PBS control group, the G0, G3, and G5 groups were selected for intestinal microbiota diversity analysis after *Aeromonas hydrophila* infection (Figure 6). Compared with the PBS control group, the endemic communities of the G0, G3, and G5 groups were two, zero, and one, respectively (Figure 6A). In addition, the Shannon and Simpson indices showed that the diversity of the intestinal microbiota was reduced after infection (Figure 6E,F). After PCA, there was a certain degree of dispersion between the G0 group community and other groups (Figure 6B). By comparing species composition at the phylum level between groups, we found that the abundance of *Proteobacteria* and *Fusobacteria* in the intestinal microbiota was mainly altered after *Aeromonas hydrophila* infection (Figure 6C). Further analysis revealed that the main changes at the genus level were the abundance of *Cetobacterium*, *Vibrio*, *norank_f_Barnesiellaceae*, *ZOR0006*, and *unclassified_f_Erysipelotrichaceae* (Figure 6D). Analysis of the dominant intestinal microbiota revealed significant differences between *Fusobacteria* (Figure 6G) and *Cetobacterium* (Figure 6H) and this difference was the greatest in the G3 group (*p* < 0.01). We performed COG functional predictions of postinfection intestinal microbiota, and found that the functional classification did not change, but the abundance decreased significantly (Figure 6I).

## 4. Discussion

The intestinal mucosa is the body’s first barrier against external invasion and is a key element directly related to intestinal homeostasis [34]. By observing intestinal morphology, the intestinal mucosal barrier function can be ascertained [35]. The metabolism of nutrients in the intestine occurs through the action of intestinal villi, which have the function of resisting pathogen infection [36]. Previous studies have shown that a normal intestinal tract in the initial stages of development exerts a considerable effect on the mature stage and that the development of the intestine can be modulated through dietary compositions [37,38]. Changes in basal nutrient composition were first used to study the effects of dietary composition on intestinal development. A modest reduction in protein content was shown to increase the abundance of intestinal villi in weaned piglets [39]. *Megalobrama amblycephala* alters intestinal function in response to the dietary components of carbohydrate and fat [40]. Natural ingredients and formulations with bioactive components, such as growth-promoting and antibacterial components have also been developed as dietary supplements [41]. Dietary insect meal has been proven to improve intestinal morphology in free-range chickens [42]. Probiotics are also often used to improve intestinal function [43]. Antimicrobial peptides are used in many fields, such as in food and medical treatment [44]. For example, AMPs were used as microadditives to increase *Cyprinus carpio* growth and immune-related gene expression [45] and have significant effects in improving antioxidant and intestinal function [46]. In our study, we found that the addition of AMPs improved intestinal development, promoted villus growth, and increased the abundance of goblet cells (Appendix A). Previous studies have shown that goblet cells are the main producers of mucus and are involved in sensing the intestinal lumen and controlling the functioning of the intestinal immune system [47]. Goblet cells secrete intestinal mucus to protect the intestine when invaded by pathogens [48]. After *Aeromonas hydrophila* infection, we found that the intestinal goblet cells in the G0 group showed abnormal proliferation compared with those in the AMP-supplemented group (Appendix A). Studies have shown that intestinal damage leads to intestinal mucus secretion and that mucus secretion is positively correlated with the degree of intestinal damage [49]. This suggests that AMPs can reduce the damage caused by *Aeromonas hydrophila* infection, which is consistent with previous results showing that *L. acidophilus* reduces intestinal damage caused by *Salmonella* infection [50]. We also contrasted the number of intestinal goblet cells before and after an *A. hydrophila* infection (Appendix A). The findings revealed that whereas intestinal goblet cell numbers in most experimental groups did not significantly increase, they considerably increased in the G0 group. As a result, it is possible to hypothesize that the injection of AMPs enhances the gut’s innate immunity while also significantly boosting intestinal disease resistance following *A. hydrophila* infection.

Alteration of intestinal redox status is an intrinsic key factor in regulating the state of intestinal tissue, and the overproduction of ROS is one of the main culprits leading to inflammation [51]. Altered antioxidant enzyme activities, such as glutathione (GSH), SOD, CAT, and MDA are often used as evaluation criteria to detect the antioxidant function of organisms [52]. When compared to the G0 group, the G3 and other groups had considerably higher T-AOC and SOD enzymatic activity (Appendix A). Simultaneously, the relative activity of MDA, a key lipid peroxidation product, was suppressed and dramatically reduced in the G3 group. It is worth noting that an appropriate supplementation amount is key to ensuring optimal AMP function, as excessive addition will reduce the activity of antioxidant enzymes and lead to the development of inflammation. This is consistent with previous research on *Epinephelus coioides*. The function of intestinal nutrition absorption is represented in the activity of intestinal digestive enzymes as well as the shape of intestinal villi [53]. By measuring the activity of α-chmo, α-Ams, and Lip, we found that the intestinal digestive enzymes were significantly activated, and that the G3 group had the most obvious effect (Appendix A). This indicates that AMPs improve intestinal digestive function not only by promoting the development of intestinal villi but also by increasing the activity of intestinal digestive enzymes. This result confirms the conclusion that AMPs improve intestinal digestive enzymes in studies of broiler chickens and juvenile goats [54].

Intestinal oxidative stress is thought to be a side effect of the inflammatory process, with redox imbalance being a symptom of inflammatory assaults. Intestinal inflammation is associated with the activation of NF-κB signaling factors in the intestinal mucosa, which is regulated by the toll-like receptor (TLR) signaling pathway [47]. Therefore, we assessed the levels of immunomodulatory factors (*TLR-4* and *MYD88*) and inflammatory factors (*TNF-α, IL-10* and *IL-11*) in the intestine. The results showed that AMPs raised the levels of *TLR-4* and *MYD88*, upregulated the levels of the proinflammatory factor *TNF-α* and downregulated the levels of the anti-inflammatory factors *IL-10* and *IL-11*. Studies have shown that the production of inflammatory cytokines can improve innate immunity and resistance to pathogen invasion [55]. After infection with *Aeromonas hydrophila*, we found that the *IL-10* and *IL-11* expression levels were upregulated in each group, while *TLR-4*, *MYD88*, and *TNF-α* were down regulated. The AMP-supplemented groups showed a greater change in expression than the G0 group. Research on antimicrobial peptides has shown that they inhibit the TLR signaling pathway and regulate the levels of inflammatory chemokines (*TNF-α*, *IL-1β*, etc.), while promoting the levels of anti-inflammatory factors (*IL-10*, *IL-11*, etc.), representing an anti-inflammatory mechanism of antimicrobial peptides [56]. Therefore, our results suggest that the addition of AMPs enhances immunity and anti-inflammatory function.

The intestinal microbiota aids the host immune system, improves intestinal barrier function, and prevents colonization of the intestine by pathogenic bacteria [57]. In the alpha diversity study, the Shannon, Simpson, Chao1, and Ace indices were utilized to express the diversity and richness of species in the sample. The Shannon, Ace, and Chao1 indices were all higher in the AMP-supplemented groups, but the Simpson index was lower, which was consistent with prior research [21]. The distribution of the intestinal microbiota of fish was dominated by Fusobacterium and Proteobacteria, while Firmicutes, Bacteroidetes and Actinomycetes were also observed as part of the main microbiota [46]. Our results show that AMPs boosted the relative abundance of *Firmicutes* and *Bacteroidetes* while lowering the abundance of *Fusobacteria*. *Firmicutes* and *Bacteroidetes*, which are obligate anaerobic bacteria, dominate bacterial communities in a healthy intestine. In contrast, intestinal inflammation is associated with elevated facultative anaerobic bacteria such as *Proteobacteria* [58]. *Bacteroidetes* and *Firmicutes* have key roles in intestinal energy metabolism, whereas an increase in the *Firmicutes/Bacteroidetes* ratio can lead to metabolic imbalances [59]. Our results imply that adding AMPs boosted *Bacteroidetes* and *Firmicutes* abundance while decreasing the *Firmicutes/Bacteroidetes* ratio, thus promoting intestinal energy metabolism. The analysis of the differences between groups showed that the G3 and G0 groups had the most different species. We found significantly decreased *Fusobacteria* and significantly increased *Actinobacteria* numbers at the phylum level, and significantly decreased *Cetobacterium* and significantly increased *Flavobacterium* and *Rhodococcus* numbers at the genus level. *Actinobacteria* are a minority of bacteria in the intestinal flora but they are critical for the formation and maintenance of intestinal homeostasis [60]. In particular, *Cetobacterium* is the main species that constitutes *Fusobacterium* in the fish microbiota [61]. SCFAs, metabolites of *Cetobacterium*, play a key function in maintaining the integrity of the intestinal barrier by producing mucus and preventing inflammation [62]. This claim was also verified in subsequent infection experiments. *Aeromonas hydrophila* infection resulted in a considerable rise in both *Fusobacterium* and *Cetobacterium,* as well as a massive reduction in *Proteobacteria*. A large increase in *Cetobacterium* can effectively alleviate intestinal inflammation and protect intestinal barrier function, which is consistent with our results on the intestinal anti-inflammatory factors IL-10 and IL-11 (Figure 5D,E). The abundance of *Proteobacteria* decreased in all groups, but the decrease in the G3 group was more significant than the G0 group, which was consistent with our abovementioned statement that the count of *Proteobacteria* was proportional to the level of proinflammatory factors.

Previous research has shown that AMPs influence the gut and change the composition of the gut microbiota. The expression of intestinal immune factors is closely related to the abundance of beneficial bacteria, indicating that Proteobacteria, Verrucomicrobia and Nitrospirae can suppress inflammation and improve intestinal immunity. The inhibition of the diversity of intestinal bacteria is linked to the fact that high dosages of AMPs eventually have a suppressive impact on intestinal immunity. It is clear from the pairwise analysis at the OTU level that the pattern with the addition of AMPs was one of first growing and then decreasing. AMPs have a broad spectrum of activity. We hypothesize that high concentrations of AMPs may influence good bacteria while inhibiting bad bacteria. Following *Aeromonas hydrophila* infection, the analysis of the gut’s microbial diversity revealed that the G5 group had the greatest diversity. This further establishes the possibility that the dose may be positively correlated with the inhibitory action of AMPs on dangerous bacteria. However, the potent antibacterial effects of high-dose AMPs will have a detrimental effect on the uninfected gut, perhaps reducing both hazardous and beneficial bacteria.

In summary, we can boldly infer that AMPs can directly and indirectly improve intestinal function. AMPs can improve innate immunity by activating the TLR signaling pathway and regulating the expression of immune-related genes. They can also maintain intestinal homeostasis and intestinal functional integrity by regulating the activity of intestinal antioxidant enzymes through the inflammatory response. At the same time, AMPs can improve the composition of intestinal microbiota, increase the abundance of *Firmicutes* and *Bacteroidetes* to promote intestinal energy absorption, and increase the abundance of *Actinomycetes* to maintain intestinal homeostasis. When infected by pathogens, AMPs can inhibit the TLR pathway and promote the expression of anti-inflammatory factors through their unique anti-inflammatory mechanism. Moreover, AMPs can endow the intestinal microbiota with a more favorable species composition after infection with pathogenic bacteria. The significantly increased *Cetobacterium* and the significantly decreased *Proteobacteria* both contributed to an organism’s ability to acquire higher disease resistance through SCFAs and anti-inflammatory factors. It should be reiterated that the supplementation level of AMPs is very important because high levels reduce their effect and can even have adverse effects.

## 5. Conclusions

In conclusion, adding an appropriate amount of AMPs (400 mg/kg in this experiment) could directly and indirectly improve the intestinal function of *Carassius auratus* var. *Pengze* and regulate the composition of intestinal microbes. In addition, AMPs can provide direct antibacterial effects when organisms are infected with pathogens and indirectly improve intestinal disease resistance through a more favorable intestinal microbial structure. However, at a high dose, the beneficial effect of AMPs is inhibited and this observation requires more research. This study also revealed the potential use of AMPs as dietary supplements and antibiotic substitutes. 

## Figures and Tables

**Figure 1 antioxidants-11-01756-f001:**
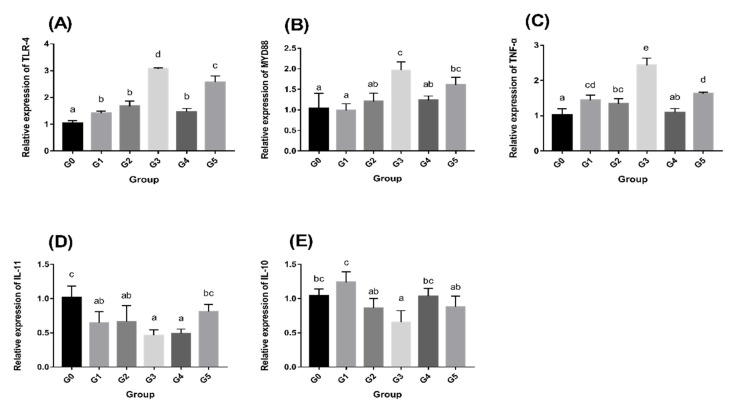
Intestinal immune-related gene expression levels after the addition of AMPs. (**A**) Relative expression of *TLR-4*; (**B**) Relative expression of *MYD88*; (**C**) Relative expression of *TNF-α*; (**D**) Relative expression of *IL-11*; and (**E**) Relative expression of *IL-10*. The values represent the means with standard errors (*n* = 3). Values which do not have a common superscript differ significantly (*p* < 0.05).

**Figure 2 antioxidants-11-01756-f002:**
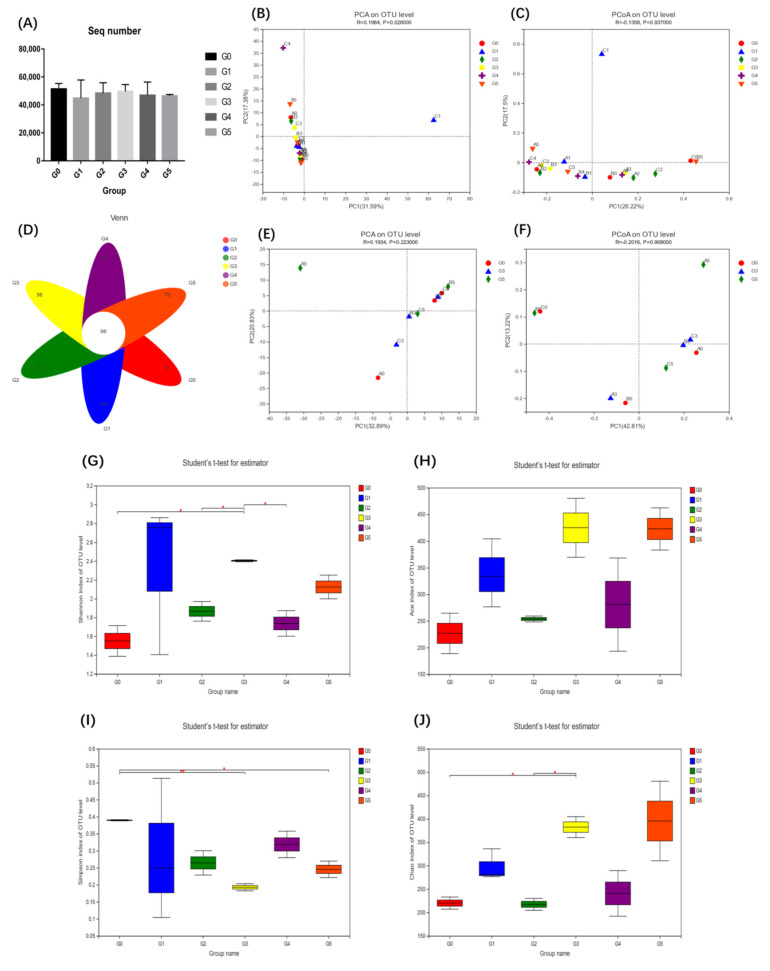
Comparison of the intestinal community composition, diversity, and richness between all groups. (**A**): the sequence count; (**B**,**C**): the PCA and PCoA analysis at the OTU level; (**D**): the OTU overlap analysis; (**E**,**F**): the PCA and PCoA analysis of G0, G3, and G5 groups; (**G**): Shannon index; (**H**): Ace index; (**I**): Simpson index; and (**J**): Chao1 index. *n* = 3 per group. * *p* < 0.05, ** *p* < 0.01.

**Figure 3 antioxidants-11-01756-f003:**
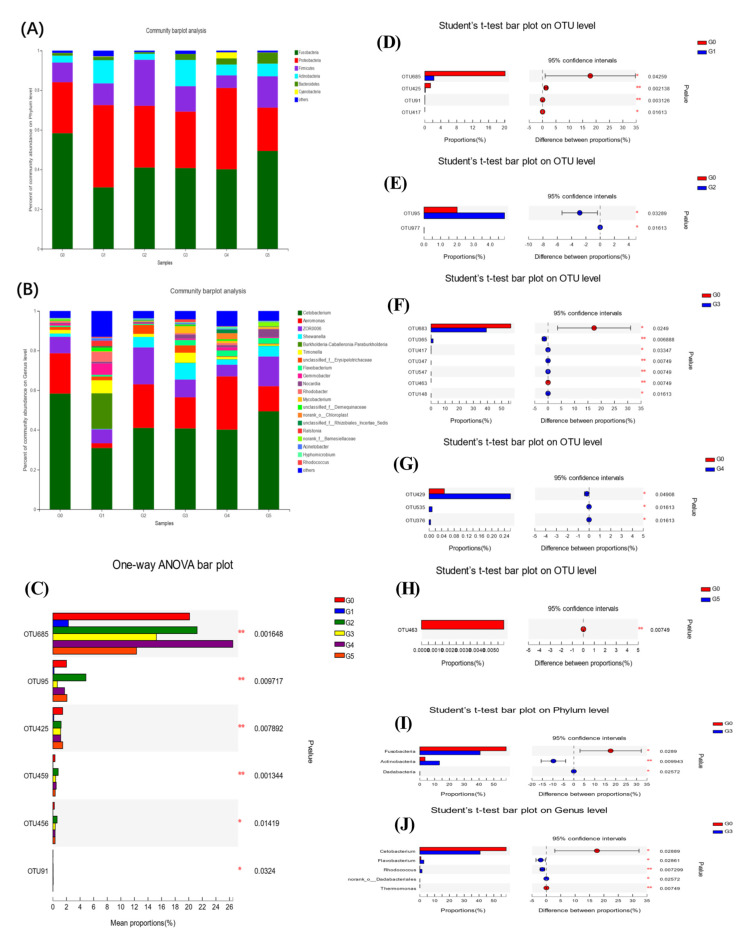
Comparison of the beneficial intestinal microbiota and community differences between all groups. (**A**): the phylum level; (**B**): the genus level; (**C**): the significantly different community between groups at the OTU level; (**D**–**H**): pairwise comparison at the OTU level; (**I**): phylum with significant difference between G3 and G0 groups; and (**J**): genus with significant difference between G3 and G0 groups. *n* = 3 per group. * *p* < 0.05, ** *p* < 0.01.

**Figure 4 antioxidants-11-01756-f004:**
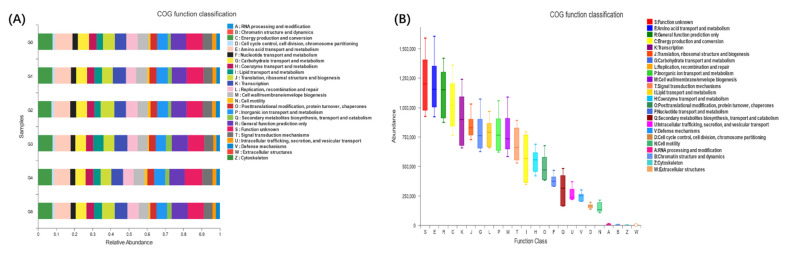
Comparison of the functional classification of COG between all groups. (**A**): COG functional abundance; and (**B**): COG functional relative abundance. *n* = 3 per group.

**Figure 5 antioxidants-11-01756-f005:**
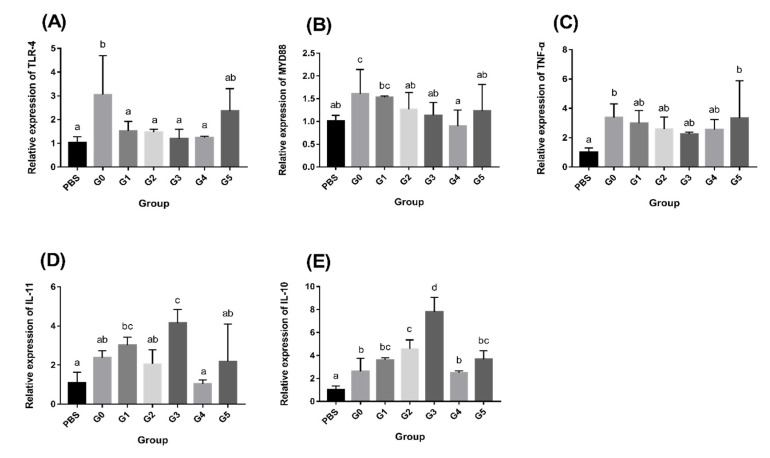
Expression of intestinal immunity-related genes following *Aeromonas hydrophila* infection. (**A**) Relative expression of *TLR-4*; (**B**) Relative expression of *MYD88*; (**C**) Relative expression of *TNF-α*; (**D**) Relative expression of *IL-11*; and (**E**) Relative expression of *IL-10*. The values represent the means with standard errors (*n* = 3). Values which do not have a common superscript differ significantly (*p* < 0.05).

**Figure 6 antioxidants-11-01756-f006:**
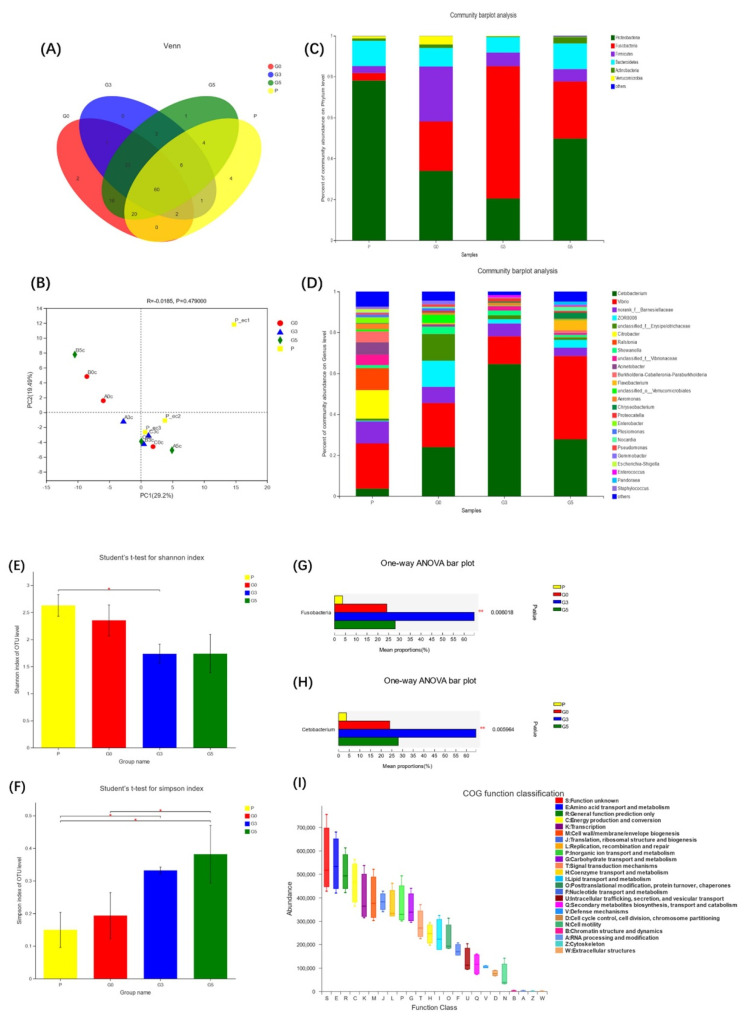
Intestinal microbiota changes after *Aeromonas hydrophila* infection between all groups. (**A**): OTU overlap analysis; (**B**): PCA at the OTU level; (**C**,**D**): community differences at the phylum and genus levels; (**E**): index of Shannon; (**F**): index of Simpson; (**G**,**H**): significant difference at the phylum and genus levels; and (**I**): relative abundance of COG functions. *n* = 3 per group. * *p* < 0.05, ** *p* < 0.01.

**Table 1 antioxidants-11-01756-t001:** Feedstuffs and experimental diets in this study.

Ingredient (%)	G0	G1	G2	G3	G4	G5
Canola meal	20.00	20.00	20.00	20.00	20.00	20.00
Soybean meal	35.00	35.00	35.00	35.00	35.00	35.00
Wheat flour	12.00	12.00	12.00	12.00	12.00	12.00
Corn starch	5.00	5.00	5.00	5.00	5.00	5.00
Fish meal	15.00	15.00	15.00	15.00	15.00	15.00
Mixed antimicrobial peptide	0.00	0.01	0.02	0.04	0.08	0.16
Calcium dihydrogen phosphate	2.00	2.00	2.00	2.00	2.00	2.00
Fish oil	3.00	3.00	3.00	3.00	3.00	3.00
Soybean oil	3.00	3.00	3.00	3.00	3.00	3.00
Premix *	1.00	1.00	1.00	1.00	1.00	1.00
Microcrystalline cellulose	3.20	3.19	3.18	3.16	3.12	3.04
Carboxymethyl cellulose(CMC)	0.50	0.50	0.50	0.50	0.50	0.50
Choline chloride (50%)	0.20	0.20	0.20	0.20	0.20	0.20
Vitamin C phosphate	0.10	0.10	0.10	0.10	0.10	0.10
Proximate composition						
Moisture	13.42	12.95	13.16	13.34	13.08	13.12
Crude protein	36.10	35.98	36.22	36.16	35.88	36.16
Crude lipid	8.41	8.23	8.37	8.31	8.46	8.29
Ash	7.24	7.45	7.38	7.55	7.35	7.51

* Premix reference to previous studies [26].

**Table 2 antioxidants-11-01756-t002:** Primer sequences for RT–qPCR.

Primer Name	Sequence (5′–3′)	Size of PCR Amplicon (bp)
*TLR-4*-F	GTAGTTCTTTTGTCATTCTTGGTT	122
*TLR-4*-R	TGACCCAATCTTCATCATAGC	
*TNF-α*-F	CGCGACTGACACTGAAGACC	79
*TNF-α*-R	GCAGGAGTTCTGTGGTGGTG	
*MYD88*-F	TGACAGCCTACACCCTT	166
*MYD88*-R	GATGCCGTGGCGACTA	
*IL-11*-F	CCACAGAGATTGATCACCATAGG	191
*IL-11*-R	TGTCAGCTTTGGTACTGAGC	
*IL-10*-F	GTTATTAAAGCCATGGGAGAGC	198
*IL-10*-R	GAAGTCCATTTGTGCCATATCC	
*β-actin*-F	CTCCCCTCAATCCCAAAGCCAA	127
*β-actin*-R	ACACCATCACCAGAATCCATCA	

**Table 3 antioxidants-11-01756-t003:** AMPs affect intestinal enzyme activity.

Items	G0	G1	G2	G3	G4	G5
T-AOC	59.33 ± 6.47 ^a^	113.15 ± 2.99 ^b^	67.35 ± 15.34 ^a^	129.91 ± 23.10 ^b^	69.59 ± 7.57 ^a^	46.91 ± 2.79 ^a^
SOD	48.72 ± 6.79 ^ab^	64.00 ± 10.66 ^ab^	38.23 ± 2.84 ^a^	118.73 ± 48.25 ^c^	84.24 ± 14.64 ^bc^	62.36 ± 21.58 ^ab^
MDA	14.64 ± 0.52 ^b^	14.61 ± 0.78 ^b^	13.23 ± 0.25 ^ab^	10.54 ± 1.57 ^a^	14.66 ± 0.36 ^b^	15.01 ± 3.30 ^b^
α-Ams	0.11 ± 0.02 ^ab^	0.17 ± 0.03 ^b^	0.30 ± 0.09 ^c^	0.40 ± 0.07 ^c^	0.07 ± 0.01 ^a^	0.14 ± 0.06 ^ab^
α-chmo	1.08 ± 0.44 ^a^	4.44 ± 0.64 ^bc^	3.72 ± 1.13 ^bc^	7.79 ± 2.20 ^d^	5.86 ± 1.01 ^cd^	2.69 ± 0.73 ^ab^
Lip	1.51 ± 0.22 ^a^	3.38 ± 0.70 ^cd^	2.89 ± 1.30 ^bc^	4.20 ± 0.31 ^d^	1.85 ± 0.18 ^ab^	1.51 ± 0.51 ^a^

The values represent the means with standard errors (*n* = 3). Values which do not have a common superscript differ significantly (*p* < 0.05).

**Table 4 antioxidants-11-01756-t004:** The vitro bacteriostatic activity of AMPs.

Items	0 (65 % Ethanol)	1 (160 mg/mL)	2 (16 mg/mL)	3 (1.6 mg/mL)	4 (0.16 mg/mL)	5 (0.016 mg/mL)
Inhibition zone diameter (mm)	0	5.53 ± 1.05	4.78 ± 0.65	4.68 ± 0.48	0	0

## Data Availability

The authors declare that the data supporting the findings of this study are available within the article and its Appendix A.

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
