# Peer review of "Dietary Antimicrobial Peptides Improve Intestinal Function, Microbial Composition and Oxidative Stress Induced by Aeromonas hydrophila in Pengze Crucian Carp (Carassius auratus var. Pengze)"

_antioxidants, 2022, doi:10.3390/antiox11091756_

Round 1

Reviewer 1 Report

There is growing evidence that phagocytic cells express AMPS as a first line immune defense.Piscidins and other AMPS interact and induce immune cell chemotaxis. The nature of this interactome is not fully investigated, nor the localization of immune cells in the intestinal epithelium of fishes and the immune cells phenotypes that play an important role in the innate immunity and homeostasis of the fish gut epithelium.AMPS are defense molecules that directly kill bacteria, and interact with a broad range of molecules to the formation of secondary structures that are conducive to membrane disruption or translocation and ultimatately cell death.

The present study is a valid contribution to the data in literature showing AMPS supplementation is a key to understand they contribute to the improvement of the  intestinal function and pathogen resistance in the species investigated, but the authors do not provide sufficient and detailed results how the immune cells of the intestinal epithelium ie mast cells and macrophages express AMPS, and their importance as key regulators of intestinal function and gut homeostasis. Also the primary function of the gut epithelium is the transport fluid and the electrolytes from the lumen.Absorption predominates and is necessary for water transport. Central to the regulation of water transporti s the acetylcholine, the predominant transmitter of the enteric nervous system and the function of the immune cells that engulg mucous goblet cells regulating their proliferation including the GAG-AMP interaction on the surface intestinal epithelium to attract chemoattractants, including chemokine- like AMPS.

I think this study and others in this emerging field of research, can be revised in the light of the literature data suggested above, and mainly point to the interaction of immune cell roles in intestinal morphology and function in fish.

Also I found the data presented fit the scope of more concerned journals ie. “ Cells “ and “ Biology “ rather than “ Antioxidants “.

Reviewer 2 Report

In this study, Wang and collaborators provide evidence that dietary supplementation with mixed antimicrobial peptides can improve intestinal function, antioxidant activity, and microbial composition in Pengze crucian carp. 

The paper extended previous findings (Wang et al., 2021; Fish and Shellfish Immunology 114 (2021) 112–118) in which the Authors, by using the same experimental design, reported that AMPs improve growth performance, antioxidant and immune responses, and disease resistance in other tissues of same species. In some cases, i.e. antioxidant activity and immune-related gene expression, results are analogous. 

The question addressed by this work is of interest, the paper is well written and the experimental design is appropriate. However, some remarks, raised below, need some point of comment/discussion: 

-   In the Discussion it is reported: “In our study, we found that the addition of AMPs improved intestinal development, promoted villus growth, and increased the abundance of goblet cells (Fig. S1). Moreover, it is also reported that: “After Aeromonas hydrophila infection, we found that the intestinal goblet cells in the G0 group showed abnormal proliferation compared with those in the AMPs-supplemented group (S2)”. Can the Authors clarify because AMPs improve goblet abundance also in absence of infection? And if this is the case, is there significant difference in the goblet abundance before and after A. hydrofila infection? 

-   Table S3: fish selected from the G0 group and injected with PBS served as the control group in A. hydrophila infection analysis; how the increased intestinal goblet cell abundance in the G0 group compared to the PBS control group can be explained? 

-   How was the dose of A. hydrophila chosen?

-  Did the authors check the effectiveness of AMPs supplementation against higher doses of A. hydrophila?

-   A tentative explanation/hypothesis for the missing effect of higher AMPs doses should be provided.

-   Are details regarding the class of AMPs (namely C and P) used available?

Reviewer 3 Report

This manuscript, “Dietary Antimicrobial Peptides Improve Intestinal Function……” by Wang et al., reports the benefits of using Anti-microbial peptides. This study is a continuation of the author’s previous publication in Fish and Shellfish Immunology 114 (2021) 112–118. The current study is well designed and executed with necessary and appropriate controls. The study focused on intestinal health and measured the changes using histology, biochemistry, and microbial diversity using sequence analysis. This study suggests that using appropriate levels of AMP (400mg/kg) is beneficial, with caution that an overdose of AMPs could lead to adverse effects. The manuscript is very well written with appropriate references. 

I would recommend the manuscript for publication after a minor revision. I have the following questions and suggestions.

1.     What is the composition of the antimicrobial peptide authors used in this study? 

2.     Are the AMPs produced in-house by authors? How is it controlled for purity?

3.     Since this study is a continuation of the previous publication by the authors (Fish and Shellfish Immunology 114 (2021) 112–118), why did the authors not look specifically at more immune effectors in the intestine? It looks like the TLR4 and MYD88 regulation seems to be a general response to AMPs rather specific in the intestine. 

4.     While the microbiome analysis suggests that the enriched functions mainly included

energy production and conversion, amino acid transport, and metabolism, why were the FBW, WGR, and SGR of the G3 group significantly lower than those of the G0? According to your previous publication Fish and Shellfish Immunology 114 (2021) 112–118. Please clarify.

5.     I am a bit surprised how narrow the concentration (400mg/kg) is in group 3 because the benefit of AMP treatment seems to fade after this concentration. Also, this concentration affects the growth rate of fish. Looks like this concentration of switches the metabolism towards defense response rather than growth. However, the microbiome diversity contradicts the above statement. The authors need to see the two studies (current and previous studies) together and interpret the data.

6.     Further, did the authors validate the antibacterial activity of the AMP-C and AMP-P invitro against Aeromonas? If not, I strongly suggest the authors include that data.  

Round 2

Reviewer 1 Report

Although the Authors have provided a good model for a different field of study,they have emphasized in their rebuttal letter the regulatory roles of AMPs in the secretion of immunoregulatory substances by the gut immune cells, and so pointed out the existing gaps in need of further study.